# Interpreting Iron Homeostasis in Congenital and Acquired Disorders

**DOI:** 10.3390/ph16030329

**Published:** 2023-02-21

**Authors:** Natalia Scaramellini, Dania Fischer, Anand R. Agarvas, Irene Motta, Martina U. Muckenthaler, Christina Mertens

**Affiliations:** 1Department of Clinical Sciences and Community Health, University of Milan, 20122 Milano, Italy; 2Unit of Medicine and Metabolic Disease, Fondazione IRCCS Ca’ Granda Ospedale Maggiore Policlinico, 20122 Milan, Italy; 3Department of Anesthesiology, Heidelberg University Hospital, Im Neuenheimer Feld 420, 69120 Heidelberg, Germany; 4Center for Translational Biomedical Iron Research, Department of Pediatric Oncology, Immunology, and Hematology, University of Heidelberg, INF 350, 69120 Heidelberg, Germany; 5Molecular Medicine Partnership Unit, 69120 Heidelberg, Germany; 6DZHK (German Centre for Cardiovascular Research), Partner Side, 69120 Heidelberg, Germany

**Keywords:** iron, iron metabolism, rare disease, hematology, anemia, iron overload, iron deficiency, hemochromatosis

## Abstract

Mammalian cells require iron to satisfy their metabolic needs and to accomplish specialized functions, such as hematopoiesis, mitochondrial biogenesis, energy metabolism, or oxygen transport. Iron homeostasis is balanced by the interplay of proteins responsible for iron import, storage, and export. A misbalance of iron homeostasis may cause either iron deficiencies or iron overload diseases. The clinical work-up of iron dysregulation is highly important, as severe symptoms and pathologies may arise. Treating iron overload or iron deficiency is important to avoid cellular damage and severe symptoms and improve patient outcomes. The impressive progress made in the past years in understanding mechanisms that maintain iron homeostasis has already changed clinical practice for treating iron-related diseases and is expected to improve patient management even further in the future.

## 1. Introduction

### 1.1. Systemic Iron Metabolism

#### 1.1.1. Iron Uptake and Release

Iron is an essential element needed for systemic and cellular functions. However, in excess, iron is toxic. Since there is no physiologic excretory mechanism for excess iron, uptake, storage, and release are highly regulated processes (Figure 1). Most of the body’s iron is located in the hemoglobin of erythrocytes (~2–4 g), as it is needed for oxygen transport to tissues [1]. In tissues, iron is incorporated into muscle myoglobin or metalloproteins (~300 mg females/~1000 mg males). Under physiological conditions, most of the iron is stored in ferritin or hemosiderin in the liver, spleen, and bone marrow [2]. Iron enters the body from the diet via duodenal enterocytes (1–2 mg/day). Under aerobic conditions, iron is oxidized to ferric iron (Fe^+3^); therefore, dietary iron first needs to be reduced in the duodenal lumen to ferrous iron (Fe^+2^) before it can be absorbed [3] (Figure 1). This step is facilitated by the ferrireductase duodenal cytochrome b (DCYTB), followed by the transport through the apical membrane of the duodenal enterocytes via divalent metal transporter 1 (DMT1) [4]. The release of iron to the plasma occurs at the basolateral membrane of duodenal enterocytes via the iron exporter ferroportin [1]. The ferroxidase hephaestin (Heph) assists in the release of iron by oxidizing it from the ferrous to the ferric state [5]. The soluble homolog ceruloplasmin (CP) mediates the oxidation of ferrous iron released by macrophages via ferroportin, facilitating the binding to the plasma iron carrier protein transferrin (Tf) [3]. In addition, CP is believed to also assist in the oxidation of iron from enterocytes under conditions of high iron demand. Only small amounts of iron circulate in the bloodstream, where it is bound to Tf (4 mg) [6]. The amount of Tf-bound iron is highly dynamic and has a turnover rate of ~10 times/day to supply iron for erythropoiesis. In contrast to the small amount of iron that is taken up via duodenal enterocytes, erythropoiesis needs ~20–25 mg of iron per day [7]. Most of the iron required for erythropoiesis is supplied by reticuloendothelial macrophages that phagocyte aged red blood cells [1]. Red blood cells have a half-life of around ~120 days, and their aging drives the externalization of phosphatidylserine, the peroxidation of membrane-bound lipoproteins, the loss of sialic acid residues, and the formation of senescence neoantigens [8]. These modifications are recognized by macrophages located in the liver and spleen that, in turn, phagocytize the senescent or aged erythrocytes by a receptor-mediated process. Efficient iron recycling is profoundly challenged under conditions of allogenic blood transfusions or hemolysis. Hemolysis occurs in response to sepsis, pregnancy complications (preeclampsia), genetic disease (e.g., sickle cell disease), blood transfusions, or the application of medical devices [9,10,11].

#### 1.1.2. Iron Handling in Diseases Associated with Hemolysis

Hemolysis occurs both in the bloodstream (intravascular hemolysis) and in the spleen (extravascular hemolysis), both in health and disease and during medical interventions such as extracorporeal circulation (ECMO) [12]. Excessive free hemoglobin, heme, and iron in the bloodstream may exert toxic effects by acting as erythrocyte damage-associated molecular patterns (DAMPs), by scavenging nitric oxide (NO), and by promoting reactive oxygen species (ROS) production and hemolysis-mediated toxicity, which can result in end-organ damage such as acute kidney injury; it is important that they are cleared from the circulation rapidly. Macrophages play an important role in hemolysis by clearing red blood cells and their derivates. The release of heme and labile iron catalyzes the production of reactive oxygen species (ROS). Hemolytic RBCs have to be rapidly cleared from the circulation by reticuloendothelial macrophages in the spleen and liver [13] to prevent intravascular hemolysis and the release of toxic heme and iron in the circulation.

During intravascular hemolysis, for example, the heme and hemoglobin that exceed the binding capacity of scavengers, such as hemopexin (Hpx) and haptoglobin (Hp), will be internalized by macrophages and trigger polarization towards a macrophage phenotype that produces high levels of inflammatory cytokines [14]. Hemoglobin is recognized as bound to Hp via the cluster of differentiation (CD)163 on the macrophage surface [15]. The oxidation of hemoglobin converts heme-bound ferrous iron to ferric iron (hemin), which binds to Hpx and is recognized as an Hpx–heme complex by the CD91 receptor. After internalization, the iron is recycled via the metabolic activity of the membrane-bound enzyme hemeoxygenase-1 (HO-1) [16,17] into iron and bilirubin, whereby the iron is exported from macrophages via ferroportin and loaded onto transferrin to supply iron for erythropoiesis. Under conditions where the Tf-binding capacity is exceeded, toxic non-transferrin-bound iron (NTBI) is produced [18,19] which is known to cause tissue damage [14,20,21]. In addition, heme has been shown to directly bind to toll-like receptors (TLR)4 and to trigger pro-inflammatory signals [22,23].

#### 1.1.3. Adjustment of Iron Hemostasis According to Cellular Needs

Iron homeostasis is controlled not only by iron absorption, trafficking, and cellular iron uptake, but it is also balanced according to utilization and storage. Therefore, the expression of ferroportin, ferritin, and TfR1, which are essential for the uptake of Tf-bound iron, is post-transcriptionally regulated by iron-regulatory proteins 1 and 2 (IRP1 and 2) binding to iron-responsive elements (IREs) within the untranslated regions of their mRNAs [24]. In iron-deficient cells, the mRNA is stabilized by the binding of IRPs to IREs in the 3′ untranslated region-UTR, and ferroportin is translationally repressed by IRP-binding in the 5′ UTR. This allows iron accumulation in the cell. In iron-repleted cells, IRPs are inactive as RNA-binding proteins, and the degradation of TfR1 mRNA occurs. The IRE/IRP system also accounts for the regulation of further proteins directly or indirectly involved in iron metabolism [3,25]. 

Hypoxia is tightly coupled to the control of iron metabolism. In the hypoxic duodenal environment, IRP1 controls the translation of hypoxia-inducible factor 2α (HIF-2α) [26], which favors iron uptake and release from enterocytes via upregulation of DMT-1 and ferroportin, respectively [27]. In iron deficiency, HIF-2α increases erythropoietin synthesis in the kidney, whereby IRP1 binds HIF-2α 5′IRE and represses protein translation, limiting erythropoietin production, erythropoiesis, and iron consumption [28]. This process may serve to reduce erythropoiesis to avoid excessive iron consumption.

In the duodenum, iron is absorbed from the diet by divalent metal transporter-1 (DMT-1) before it is released into the plasma iron pool by the iron exporter ferroportin (FPN). Plasma iron is used for erythropoiesis in the bone marrow, which is regulated by erythropoietin. Senescent red blood cells are recognized by specialized macrophages in the reticuloendothelial system. After the phagocytosis of erythrocytes, iron is recycled via hemeoxygenase-1 (HO-1) and released by ferroportin during hemolysis. Hemopexin-heme and haptoglobin–hemoglobin complexes are recognized by their respective receptors cluster of differentiation (CD): CD91 and CD163. After release from cells, iron is bound to the carrier protein transferrin (Tf) or is stored in the liver. Liver and systemic iron levels are regulated by hepcidin, depending on the availability of iron. Under inflammatory conditions, hepcidin is activated and induces the sequestration of iron in iron storage sites and in the reticuloendothelial system.

### 1.2. Hormonal Regulation of Systemic and Cellular Iron Homeostasis by Hepcidin

Hormonal regulation plays an important role in fine-tuning iron availability, and it is driven by hepcidin, a liver-derived peptide hormone. Hepcidin is known as the major regulator of systemic iron availability [29]. Plasma iron levels as well as tissue iron distribution are regulated by hepcidin via inhibiting intestinal iron absorption, controlling iron recycling from macrophages and mobilizing iron from liver iron stores. Iron export via ferroportin is regulated by the binding of hepcidin, which induces endocytosis of ferroportin and lysosomal degradation or occlusion of the central cavity of ferroportin (12–15). Hepcidin is synthesized and released by hepatocytes in response to increased body iron concentrations and inflammation. Hepcidin expression is controlled by the bone morphogenic protein 2 and 6 (BMP2 and BMP6) produced by the liver sinusoidal cells [30,31]. BMP6 binds the bone morphogenic protein receptor (BMPR) and the co-receptor hemojuvelin (HJV) on the hepatocyte surface and facilitates the formation of a complex with BMP6 receptors that activate the small mother against decapentaplegic (SMAD) pathway and hepcidin transcription [32,33] (Figure 2). The detailed mechanism of Tf-bound iron sensing by hepatocytes and liver endothelial cells is still not fully understood, and additional studies are needed to understand the exact mechanism.

Circulating iron is sensed by hepatocytes: the Fe2^+^-Tf complex binds to TfR1 and mainly TfR2. Hemochromatosis protein (HFE) binds to TfR2, increasing the activation of the SMAD signaling pathway [34]. During iron deficiency, HFE is bound to TfR1 and likely interacts with ALK3, a BMP2 receptor known to regulate hepcidin expression; HFE-binding prevents the ubiquitination of ALK3 and its proteasomal degradation. Interestingly, HFE-knockout mice or mice with mutated TfR2 receptor expression fail to increase hepcidin in response to high Tf-iron loads but can increase hepcidin in response to chronic iron loading with increased liver iron stores [35]. HFE belongs to the class of major histocompatibility complex (MHC) class I–like proteins [36]. 

Hemochromatosis is a congenital disorder characterized by the accumulation of iron in the parenchymal cells of the liver, heart, and endocrine glands [37]. Hemochromatosis is characterized by a systemic iron overload due to a deficiency of hepcidin and uncontrolled iron release via ferroportin. Hemochromatosis can also be caused by mutations in the HFE gene or in other genes (collectively referred to as non-HFE hemochromatosis), that, in any case, lead to a decreased activity of hepcidin binding to ferroportin. The genes involved are HAMP (encoding hepcidin), HJV, TFR2, and PIGA, in addition to gain-of-function mutations in SLC40A1 (also known as FPN1, encoding ferroportin) [37,38].

The liver transmembrane serine protease matriptase 2, encoded by TMPRSS6, is a powerful inhibitor of hepcidin expression by cleaving the BMP co-receptor hemojuvelin [39], thereby attenuating BMP/SMAD signaling and thus decreasing hepcidin. Patients with mutations in TMPRSS6 develop iron-refractory iron deficiency anemia [21]. Similarly, TMPRSS6 deficiency in mice causes microcytic anemia and iron deficiency–related hair loss [40]. Anemia in both cases is a consequence of inappropriately high hepcidin concentrations that cause decreased dietary iron absorption [41]. 

In addition, hepcidin expression is also modulated by erythropoietic signals during anemia, hypoxia, or pathologic conditions of ineffective erythropoiesis (e.g., iron overload or inflammation). As a response, the kidney produces erythropoietin (EPO) to increase the proliferation and differentiation of red blood cells. Upon EPO binding to the EPOR in erythroblasts, erythroferrone (ERFE) is expressed, which inhibits BMP/SMAD activation of hepcidin, allowing the release of iron from storage sites to the bloodstream to be available for erythropoiesis. ERFE knockout mice are not able to suppress hepcidin in response to hemorrhage or EPO injection. The link between ERFE, hypoxia, and EPO is also demonstrated by the measurement of ERFE and EPO in healthy individuals exposed to high altitudes, where increased plasma ERFE and lower plasma hepcidin levels have been observed [42]. In contrast to iron-mediated regulation of hepcidin via the BMP-SMAD pathway, inflammation-mediated regulation occurs via the IL-6/Janus kinase (JAK)/signal transducer and activator of transcription (STAT) signaling pathways [43]. During the acute phase reaction in response to infection, interleukin (IL)-6 is produced at the site of inflammation. IL-6 activates JAK/STAT signaling, enhancing hepcidin transcription and subsequent ferroportin degradation, which in turn leads to the sequestration of iron in macrophages and hepatocytes [43]. During inflammation and infections, ferroportin expression can also be transcriptionally down-regulated via toll-like receptor (TLRs) activation [44]. Decreasing iron export from macrophages results in iron retention and hypoferremia, a mechanism termed nutritional immunity. This probably evolved to limit iron availability for invading pathogens. However, if the inflammatory state persists over extended periods, anemia from inflammation may develop as a side effect of nutritional immunity, because the reduction in iron availability not only limits pathogen metabolism and growth but also limits host erythropoiesis [1]. Hepcidin is hypothesized to be the main mediator of hypoferremia and anemia of inflammation [45]. Due to increased hepcidin levels, iron is stored in splenic and hepatic macrophages and duodenal enterocytes, limiting iron export into the plasma [46,47]. 

BMP2/6 produced by liver endothelial cells (LSECs) binds the BMP receptor (BMPR), which activates SMAD1/5/8. Hemojuvelin (HJV) acts as a BMP co-receptor. High serum iron levels and hepatic iron stores result in increased BMP6 levels and induce hepcidin mRNA transcription. SMAD4 translocates to the nucleus to bind BMP-responsive elements (BRE) in the hepcidin promoter. During iron deficiency, TMPRSS6 is stabilized on the cell surface and cleaves HJV. Thereby, the BMP6 pathway is inactivated in the absence of the ligand, suppressing hepcidin expression. Erythropoiesis is enhanced by erythropoietin signaling and large amounts of iron. High erythropoietin levels increase signaling to ERFE expression, which suppresses hepcidin transcription by sequestration of BMP6. Inflammatory cytokines such as Interleukin (IL-)6 induced by LPS signaling via toll-like receptor 4 (TLR4) stimulate macrophages to produce interleukin-6 (IL6), which activates Janus kinase 2–signal transducer and activator of transcription 3 (JAK2-STAT3) signaling to upregulate hepcidin in association with the BMP-SMAD pathway.

## 2. Iron Metabolism Imbalance

An imbalanced iron homeostasis may cause either iron deficiency or iron overload diseases. By improving our understanding of iron metabolism and iron-related parameters, the approach to the treatment of iron disorders has changed. Iron deficiencies in young females and hemochromatosis in middle-aged males have been known for centuries. Today, we can determine iron overload with iron parameters and confirm hemochromatosis by genetic testing long before iron overload and organ damage are visible. Today, we also know that iron deficiency is already a pathological entity since some tissues may have iron depletion with functional consequences before anemia develops; for example, this is well-known in cardiac settings such as heart failure [48,49]. 

Generally, iron-related diseases can be acquired or congenital, such as in the case of hemochromatosis. However, a dysregulated iron homeostasis may also be a consequence of an underlying disorder, for example, thalassemia and anemia from inflammation. Laboratory evaluation of iron status, including serum ferritin, hemoglobin, serum iron, transferrin, soluble transferrin receptors, and transferrin saturation, is fundamental for the correct diagnosis of altered iron homeostasis. Evaluation of hepcidin, even if it is not routinely available, is helpful to better define the most common iron disorders and their characteristics. 

### 2.1. Iron Overload 

Iron overload is defined by high serum ferritin (above 300 ng/L) and high transferrin saturation (Tfsat > 40%) [50]. In iron overload conditions, where hepcidin levels are low, uncontrolled iron entry into the plasma overwhelms the binding capacity of transferrin, resulting in non-transferrin-bound iron (NTBI). NTBI is a redox-active toxic form of iron, causing cellular damage and leading to organ failure in the later phases when the disease remains untreated [51]. Table 1 summarizes the most common iron parameters and hepcidin expressions for conditions of iron overload and iron deficiency. 

#### 2.1.1. Congenital Iron Overload without Anemia

Systemic forms of iron overload are characterized by increased serum ferritin and high transferrin saturation. Historically, iron overload diseases were all classified under the name of hemochromatosis. The hallmarks of hemochromatosis are hyperferremia and an iron overload in tissue parenchymal cells, with concomitant failure of tissue macrophages and intestinal enterocytes to retain iron. The standard of care for patients involves therapeutic phlebotomy, but hepcidin replacement could provide an etiologic cure [52]. With a better understanding of the pathophysiology and the genetic mechanism of hemochromatosis, it is now possible to properly reclassify them. With the name hemochromatosis (MIM #235200), we now define a genetically heterogeneous disorder in which uncontrolled intestinal iron absorption may lead to a progressive iron overload responsible for life-threatening complications. Recent advances in hemochromatosis research have highlighted that hemochromatosis is caused by mutations in at least five genes, resulting in insufficient hepcidin production, inactive hepcidin, or resistance to hepcidin binding [53]. 

Many different disease models have been generated to understand the role of iron regulatory proteins in hereditary hemochromatosis [54]. The underlying cause of the disease is a disruption in the hepcidin–ferroportin axis caused by the loss of function of the molecules involved in the activation of hepcidin synthesis, e.g., HFE, HJV, TfR1/2, or the HAMP gene itself, or mutations of the iron exporter ferroportin [48]. For example, a point mutation (C326S) in the murine Fpn locus prevents hepcidin-mediated degradation of ferroportin, resembling human hereditary hemochromatosis type 4 [55]. Mice show elevated plasma iron and ferritin levels, high transferrin saturation, and hepatic iron overload. Due to the uncontrolled release of iron via ferroportin, duodenal enterocytes and reticuloendothelial macrophages are iron-depleted [55]. Different models were also used to identify the role of HFE in the genesis of hereditary hemochromatosis [54]. Knock-out and knock-in models carrying the C282Y mutation represent the phenotypical characteristics of hereditary hemochromatosis with an iron overload. A limitation is that in the mouse model organ damage, a hallmark of human disease is not present [54]. Mice with a knockout mutation in HJV or TfR2 also represent the human form of hereditary hemochromatosis and are frequently used to investigate the molecular mechanism underlying hereditary hemochromatosis and to find new treatment options [54]. Recently, a new mutation has been described that causes severe juvenile hemochromatosis in three male children with neurologic symptoms, typical iron overload, and organ dysfunction. The mutation is in phosphatidylinositol glycan anchor biosynthesis class A (PIGA), an enzyme involved in GPI-anchor biosynthesis, which can result in improper processing of HJV and liver ceruloplasmin [38]. 

In humans, the different mutations account for different hemochromatosis phenotypes: HAMP and HJV mutations lead to a faster and more severe iron overload, with symptoms occurring in the first or second decade of life. HFE mutations cause a later onset, around the third or fourth decade of life. A revision of the hereditary hemochromatosis classification has been recently published [53] and is summarized in Table 2. The H63D variant of the HFE gene is the most common gene variant occurring in the Caucasian population [56], resulting in alterations of cellular iron homeostasis and being associated with cancer and neurological diseases [37]. Specifically, the H63D HFE genotype impacts macrophage function and activities such as proliferation [57], L-ferritin expression in response to iron loading, and secretion of BMP6 and cytokines [32,57]. Symptoms present earlier in men than women, due to the protective effect of childbirth and the menstrual cycle. For this reason, male patients have higher ferritin levels, accumulate more iron, and require earlier treatment [58]. Furthermore, one of the most common mutations, C282Y, which accounts for 80% of hereditary hemochromatosis cases, has a variable penetrance that can be influenced by environmental modifiers that worsen liver health such as alcohol consumption and fatty liver disease [59]. 

In the classical forms of hereditary hemochromatosis, the first symptoms are usually high serum ferritin levels and high transferrin saturation in combination with abnormal liver function tests. In the juvenile form, clinical findings such as diabetes, arthralgias, and liver disease are present at an early stage [52]. Genetic testing confirms the diagnosis in most cases [60]. Liver biopsies are no longer recommended, but in cases with unclear diagnoses or when genetic tests cannot explain the clinical presentation [52], liver biopsies are still needed to confirm the diagnosis. 

Treatments aim to reduce the burden of iron: weekly phlebotomy for at least one year is recommended when serum ferritin levels are above the normal range to reach a serum ferritin concentration of <100 mcg/L [61]. New therapies targeting the pathophysiological mechanism are under study, for example, treatments based on the increase of hepcidin levels via hepcidin mimetics or inducers or inhibitors of the iron export ferroportin [62].

#### 2.1.2. Congenital Iron Overload with Anemia

Inherited disorders of iron overload develop as a result of disruption of systemic iron homeostasis [63]. Alterations of genes involved in iron metabolism and transport such as DMT1 or transferrin may cause rare forms of chronic and often severe congenital anemia with iron overload and a reduction of iron available for erythropoiesis. Iron overload is poorly investigated in congenital hemolytic anemias (CHAs), a heterogeneous group of rare inherited diseases encompassing abnormalities of the erythrocyte membrane and metabolism and defects in erythropoiesis [64]. Most cases presented in the literature describe symptoms and biochemical characteristics when patients are admitted to the hospital. 

Given the small number of patients, a gap in the literature concerning the pathophysiology and possible therapeutic targets still needs to be filled. Extensive descriptions and numerous studies instead are dedicated to diseases involving mutations in globin chains associated with ineffective erythropoiesis or in genes involved in heme synthesis that can also cause iron accumulation and anemia, namely “iron loading anemias” [65]. 

##### Defect in Systemic Players of Iron Metabolism

A growing group of rare anemias is related to genetic defects in genes controlling iron metabolism. The patients usually present with hypochromic microcytic anemia associated with a visceral iron overload. The genetic defects discovered can either be related to the iron uptake by the TfR1 cycle (atransferrinemia, defects of DMT1, defects of STEAP3), mitochondrial iron utilization (X-linked sideroblastic anemia), or iron recycling (aceruloplasminemia). The prevalence of these conditions is quite rare, totaling less than one hundred patients [66]. Mutations in DMT1 (MIM #206100) cause a reduction in iron absorption and iron egress from cytosolic endosomes [67]. Patients with mutations in the DMT1 gene present, in some cases, microcytic hypochromic anemia since birth; in other cases, anemia appears in adulthood. All patients present increased transferrin saturation and high serum ferritin levels, and hepcidin is abnormally low relative to iron overload [68]. Mutations in the gene encoding transferrin (MIM #209300) cause a very rare recessive autosomal anemia due to reduced iron delivery to the bone marrow and severe iron overload. The few patients described in the literature present high levels of serum iron, very low levels of transferrin, high levels of serum ferritin, and signs of iron overload. Iron is present in circulation as NTBI [69]. A transient relief of the symptoms is possible with an infusion of apo-transferrin [70]. Aceruloplasminemia (MIM #604290), beyond neurological problems, presents severe anemia with hyperferritinemia. Iron excess is attributed to a lack of ceruloplasmin-related ferroxidase activity that, in turn, causes an upstream defect in the cellular iron export function of ferroportin. However, the absence of splenic iron overload does not favor this explanation when considering that ferroportin is particularly expressed in this tissue [71,72].

##### Non-Transfusion-Dependent Thalassemia

Thalassemias are characterized by a quantitative defect in the synthesis of globin chains and chronic anemia of variable severity characterized by a reduced mean corpuscular volume (MCV), mean corpuscular hemoglobin (MCH), and mean corpuscular hemoglobin concentration (MCHC) [73]. Hemolysis, chronic anemia, and ineffective erythropoiesis with splenomegaly, extramedullary expansion, and iron overload are the hallmarks of thalassemia, but the wide genotypical variety is accountable for a wide phenotypical presentation [74]. Clinical classification of thalassemias distinguishes transfusion-dependent thalassemia (TDT, MIM #613985, #604131) from non-transfusion-dependent thalassemia (NTDT, MIM #613978, #613985) according to the blood requirement [75]. NTDT thalassemia patients are those with chronic anemia and ineffective erythropoiesis who undergo blood transfusions only under severe conditions such as infections, pregnancy, surgery, or who are elderly [75]. The ineffective erythropoiesis and chronic anemia in NTDT account for the iron overload in such patients. The accumulation of iron from intestinal absorption in NTDT patients is slower than that observed in transfusion siderosis and may reach 3–4 mg/day or as much as 1000 mg/year [15]. Nonetheless, iron overload in NTDT patients is a cumulative process. Safe and effective chelation therapy is now available for NTDT patients ≥ 10 years, as well as for patients with a liver iron concentration ≥ 5 mg Fe/g dry weight or serum ferritin ≥ 800 ng/mL, which represent thresholds after which the risk of serious iron-related morbidity is increased [76,77].

##### Sideroblastic Anemia

Sideroblastic anemia is caused by mutations that affect mitochondrial processes involved in myelodysplastic syndromes [78]. Patients with sideroblastic anemia (SA) develop microcytic hypochromic anemia with low MCV or low/normal MCH and MCHC, associated with iron overloads, and they display peculiar erythroblasts with iron-loaded mitochondria that cluster around the nucleus [79,80]. Iron is not incorporated into hemoglobin because of a defect of heme synthesis in the iron-sulfur cluster biogenesis or in proteins involved in the detoxification and redox pathways [79]. The most common hereditary sideroblastic anemia is the x-linked SA (MIM #300751), caused by mutations of the mitochondrial enzyme delta-aminolevulinate synthase 2 (ALAS2), while recessive forms are associated with mutations in mitochondrial transport proteins. Of note, it is important to mention that some forms of SA are associated with spinocerebellar ataxia [81]. The management of SA remains primarily supportive rather than definitive: supportive care with transfusion can be considered in patients with severe and symptomatic anemia, however, it causes an aggravation of iron overload.

#### 2.1.3. Acquired Iron Overload

Chronic blood transfusions are a major cause of acquired iron accumulation. It is estimated that a red blood cell unit of around 250 mL contains approximately 200 mg of iron [82]. 

For many patients with severe chronic anemia, regular transfusions are the only therapy available and can be lifesaving [83]. Patients receiving regular RBC transfusions unavoidably and invariably develop cumulative iron overload and thereby are at risk of iron toxicity. Transfused RBCs, which have a shorter life span compared with the ~120-day life span of normal RBCs, undergo erythrophagocytosis with the release of iron that eventually overloads macrophages. When the excess iron re-enters the plasma, it surpasses the iron-carrying capacity of transferrin, circulates as NTBI, and is deposited in tissues [84]. 

An explanatory example of acquired iron overload due to transfusion is transfusion-dependent thalassemia. TDT patients present with severe chronic anemia and rely on transfusions every 2–3 weeks to maintain adequate hemoglobin levels (pre-transfusion Hb level of 9–9.5 g/dL). 

Chronic transfusion support leads to an iron overload, and hepcidin concentrations are significantly higher compared to other thalassemia forms. However, hepcidin concentrations rapidly decrease in the interval between transfusions [85]. Iron chelation is mandatory to remove or to prevent iron accumulation in different organs (liver, heart, and endocrine glands) [73,86]. Heart failure and cirrhotic liver disease are the main causes of death in TDT patients. 

The increase in systemic iron levels leads to tissue iron overload, increased serum ferritin and transferrin saturation, and, eventually, the appearance of NTBI [87]. The labile fraction of NTBI deposits in cardiac and liver tissue, fostering myocardial and liver fibrosis [88]. Liver fibrosis, which is hallmarked by continued iron deposition and inflammation, enhances the generation of reactive oxygen species and lipid peroxidation, progressing to cirrhosis [89]. Myocardial fibrosis leads to diastolic dysfunction and iron overload cardiomyopathy, characterized by arrhythmias, ventricular dilation, and severe diastolic and systolic dysfunctions [88,89].

The availability of chelators dramatically improved the survival rates of patients by preventing and reversing heart failure [90]. Deferoxamine (DFO) has been the standard iron chelator since the 1970s; it binds iron tightly, and the formed complex is excreted both in urine and stool [91]. Due to its short plasma half-life, DFO must be administered parenterally, with a long infusion, intravenously or subcutaneously; the administration route hinders the compliance of the patients. 

The development of oral chelators helped to improve adherence to therapy. The first one approved was Deferiprone (DFP), a bidentate chelator that, thanks to the absence of a net charge, can penetrate cell membranes easily and subsequently remove toxic iron from tissues; in particular, it is very efficient in removing iron from the heart [91,92,93]. Notable side effects of DFP are neutropenia and agranulocytosis, for which close monitoring by blood tests is necessary, and arthritis.

The newest oral chelator, Deferasirox (DFX), is also the most prescribed. Available in tablet and film-coated form, it is well tolerated, making it the first choice for many patients, especially those with poor adherence to DFO. Renal and liver functions must be monitored during treatment since DFX can decrease the glomerular filtration rate, increase proteinuria, and cause flares in hepatic enzymes [94]. DFX is also the only chelator approved for the treatment of iron overload in NTDT patients. In cases of severe iron overload, it was demonstrated that the combined therapy of DFP and DFO is more efficient in reducing the serum ferritin level and liver and heart iron content. Other combinations are also approved [95].

New therapies targeting iron overload are under study. Agents that stimulate hepcidin expression or activity, such as minihepcidin, TMPRSS inhibition via antisense oligonucleotides, small interference RNA, or ferroportin inhibitors [96,97,98], have been investigated in preclinical studies, demonstrating a beneficial activity. The common aim of these drugs is to alleviate anemia and prevent primary iron overload [99] by modulating iron fluxes in the bloodstream from enterocytes and macrophages. Trials on minihepcidin, however, were prematurely terminated due to safety issues and unfavorable risk–benefit ratios. Ferroportin inhibitors showed promising results in the preclinical and safety studies, and a phase 2 study will soon be initiated in NTDT patients, assessing the efficacy, safety, tolerability, pharmacokinetics, and pharmacodynamics of VIT-2763 in this patient population (ClinicalTrials.gov number NCT04364269). 

### 2.2. Iron and Cancer 

#### 2.2.1. Cancer Cell Iron Metabolism

Iron handling in the tumor microenvironment is associated with cancer progression, drug resistance, and malignant transformation [100]. Neoplastic cells are hallmarked by an increased metabolism and a higher demand for iron. Several studies have shown that iron is a driver of tumorigenesis [100]. Patients regularly donating blood generally show decreased total body iron stores and a reduced risk of cancer, while patients with iron overload disease are at an increased risk [101]. To ensure the iron supply for tumor cells, iron uptake is usually enhanced while iron export is suppressed. Tumors synthesize Tf as an autocrine mechanism to supply iron for tumor growth [102]. Malignant cells frequently overexpress TfR1, correlating with reduced patient survival [103,104]. Accordingly, inhibition of TfR1 can block tumor growth and metastasis [105]. Moreover, the upregulation of further proteins involved in iron import, such as DMT1 and DCYTB, has been reported in cancer cells [106]. By contrast, overexpression of the iron exporter ferroportin results in reduced tumor growth [107]. In addition, increased levels of hepcidin have been detected in different cancer entities, contributing to iron retention via degradation of the iron exporter ferroportin [108,109]. High BMP7 levels were linked with hepcidin overexpression in prostate cancer, especially in metastatic cancer, implicating that prostatic hepcidin might be a factor that promotes cancer cell survival [110]. Depleting intracellular iron stores with the use of iron chelators represents an attractive therapeutic opportunity, which is currently under clinical investigation [100]. 

#### 2.2.2. Innate Immune Cells as Central Iron Regulators in the TME

The tumor microenvironment (TME) plays a central role in tumor growth and progression [111]. The TME includes stromal cells, blood vessels, and immune cells such as tumor-associated macrophages (TAMs) [112]. TAMs are recruited to the TME by chemotactic molecules released by tumor cells and cancer-related stromal cells. Once recruited to the TME, macrophages differentiate into a pro-tumoral or anti-tumoral subtype, depending on the prevailing cytokines. TAMs adopt an alternative phenotype with pro-tumoral function, supporting tumor progression and invasiveness. By contrast, M1-like TAMs are tumoricidal [113,114], and iron supplementation in the form of SPIONs has been considered as an anti-tumor strategy [115,116]. Iron accumulation in TAMs was shown to be a favorable prognostic marker in patients with lung adenocarcinoma [117]. However, the M1/M2 dichotomy is a simplification considering the high macrophage plasticity that can occur in the TME. Iron metabolism is shaping the immune landscape in inflammatory/infectious diseases, also with regards to cancer-associated inflammation, one of the hallmarks of cancer [100,118]. Interestingly, cancer cells can utilize metabolic products from innate immune cells to support tumor growth and drug resistance. Specifically, macrophages and neutrophils residing in the TME can either serve as sources of iron or release factors that activate signaling pathways to control the iron metabolism in cancer cells [112]. 

TAMs are characterized by the surface expression of CD163, a hemoglobin scavenger receptor, and are therefore specialized in the uptake of heme-bound iron [119]. By the uptake and recycling of heme-bound iron these macrophages may further contribute to tumor growth and development, promoting iron release towards cancer cells via high expression of the iron exporter ferroportin and the iron-transporting protein lipocalin-2 (LCN-2) [120]. 

### 2.3. Iron Deficiency 

Iron deficiency anemia (IDA), a microcytic, hypochromic anemia (low MCV, MCHC, and MCH), is a very common condition. It is estimated that one-third of the world’s population is anemic, and half of these patients are iron deficient: this sums up to approximately two billion affected people [121]. The number of people who suffer only from iron deficiency, without the development of anemia, is underestimated. Even in the highly developed western countries, where food is available in excess, anemia is surprisingly common.

Iron deficiency includes two conditions: the total reduction of iron in the body (absolute ID) and the reduction of iron supplies to bone marrow and other organs, even if iron stores are replenished (functional iron deficiency (FID)). 

#### 2.3.1. Acquired Absolute Iron Deficiency

Absolute iron deficiency is diagnosed by low serum ferritin levels (<30 ng/mL) and low transferrin saturation (<15%) with very low hepcidin levels [121,122]. The causes of iron deficiency are an increased demand, known to occur in children during growth or in pregnant women, decreased intake, decreased absorption or malabsorption, and chronic losses. Children, women, and elderly people are severely affected all over the world by iron deficiency/iron deficiency anemia (ID/IDA). The prevalence and cause of iron deficiency differ between sexes and ages: in women of childbearing age, the most common cause of ID is menstrual blood loss. In men and women over 50 years, frequent gastrointestinal bleeding is responsible for ID/IDA [122]. During ID, hepcidin is suppressed and ferroportin is highly expressed in duodenal enterocytes, hepatocytes, and macrophages, enhancing iron uptake and release from iron stores to the plasma [34]. 

Iron restriction affects erythropoiesis in the bone marrow: the expansion of precursor RBCs and erythroblasts is reduced, and iron usage in the late stages is limited, leading to the development of anemia (IDA) [122]. 

During anemic conditions and hypoxia, the synthesis of erythropoietin is stimulated in the kidney, and hypoxia-inducible factor (HIF) levels increase. This leads to the transcription of GDF15 and ERFE, which suppress hepcidin to enhance iron uptake [123]. 

ID and IDA are treated with iron supplementation, but the treatment is effective and lasting only if the underlying causes of anemia are found and treated. Oral iron salts are the mainstay of treatment, and it has been demonstrated that administration with an alternate day schedule can improve absorption [124,125]. In iron-depleted young women, it was shown that oral iron doses given in the morning acutely increased hepcidin levels on the same day and 24 h later. This increase was strongly associated with decreased absorption from the second iron dose, given 24 h after the first. 

Frequently, patients undergoing oral iron treatment can suffer from common adverse effects due to the iron itself, such as a metallic taste, nausea, vomiting, and diarrhea, reducing their compliance and the efficacy of the treatment [126]. Indeed, patients with absorption-related issues, such as coeliac disease, especially if not well controlled, can have problems with iron salt absorption, reducing compliance and the efficacy of the treatment [127]. Liposomal iron makes use of a preparation combining phospholipids with iron delivered into target cells such as macrophages. Liposomal iron shows a high rate of gastrointestinal absorption by preventing the direct contact of iron with the gastrointestinal mucosa. It allows absorption in the small intestine and is readily available for macrophages with a low incidence of side effects. Many studies have described the usage of liposomal iron in inflammatory bowel diseases and its non-inferiority in patients with chronic kidney disease [128,129]. Intravenous iron compounds are also available and are helpful in patients who are unable to tolerate the gastrointestinal side effects of oral iron or in individuals with existing gastrointestinal disorders, which may be exacerbated by the gastrointestinal side effects of oral iron. They can also be administered in conditions in which there is a high hepcidin level, since they bypass intestinal absorption and can partially escape from hepcidin blockage [130]. 

The intravenous iron formulations which are approved are all composed of an iron salt moiety enveloped by a carbohydrate shell. Once in the bloodstream, they are taken up and processed by the reticuloendothelial system [130,131]. The concern of using IV iron, especially early preparations, such as high-molecular-weight iron dextran, is the high level of antigenicity evidenced by a high frequency of immunogenic reactions following infusions [132]. The most recent formulations, such as ferric carboxymaltose or ferumoxytol, however, show a markedly reduced level of immunogenic or anaphylactic reactions and are significantly safer with respect to the early formulations [7]. Concerning safety, the occurrence of hypophosphatemia resulting from intravenous iron treatment has been reported. Case reports describe that hypophosphatemia can occur even after a single dose [133], and clinical trials have established that the mechanism involves the bone/metabolic axis [134]. The frequency of hypophosphatemia remains unknown, given the lack of measurement of serum phosphate after treatment and that the symptoms related to hypophosphatemia (fatigue and weakness) are the same as for anemia [135].

#### 2.3.2. Acquired Functional Iron Deficiency 

Functional ID is a consequence of inflammation and is characterized by normochromic normocytic anemia with normal to increased serum ferritin levels and low transferrin values. Serum ferritin is an acute-phase protein, and its level is elevated during inflammation [136]. During systemic inflammation, high levels of inflammatory cytokines such as IL-6 enhance the expression of hepcidin, leading to a reduced expression of ferroportin and subsequent iron sequestration into macrophages, enterocytes, and hepatocytes. The aim is to reduce iron availability for Gram-negative bacteria and, as hypothesized in recent works, also prevent the production of NTBI [137]. Inflammation also affects the bone marrow, increasing the synthesis of leukocytes and myeloid precursors and reducing the synthesis of erythrocytes via the reduction of the erythropoietin stimulus [136]. Red blood cell half-life is shortened by multiple factors, in particular macrophage activation and exposure of erythrocytes to inflammatory damage. It must be considered that frequent functional iron deficiency and absolute iron deficiency can be present at the same time, but clinical suspicion and an attentive evaluation can help in the diagnosis. Indicators are serum ferritin levels between 100 ng/L and 300 ng/L, low transferrin values, and transferrin saturation < 20% [50].

Treatments that target the inflammation or infectious process can improve anemia. Erythroid stimulating agents combined with intravenous iron compounds were applied in patients suffering from anemia related to chronic kidney disease (CKD) [138]. In CKD, chronic inflammation is combined with a reduced production of erythropoietin due to renal problems; the combined therapy leads to an increase in hemoglobin levels and an improvement in anemia-related symptoms [138]. The use of this therapy in patients with anemia from inflammation who do not have CKD has not been studied, and the benefit of receiving the therapy is not yet fully understood. 

#### 2.3.3. Congenital Iron Deficiency 

##### IRIDA (Iron Refractory Iron Deficiency Anemia)

Iron refractory iron deficiency anemia (MIM #206200) is a rare recessive genetic disorder due to mutations in the TMPRSS6 gene that lead to a defect in matriptase 2, a protein involved in hepcidin regulation [139]. Without matriptase 2, ferroportin-mediated iron absorption is blocked, with a failure to absorb dietary iron despite systemic iron deficiency, as well as the coexistent failure to respond to parenteral iron [63]. The patients present with hypochromic microcytic anemia, low serum iron levels, normal to high serum ferritin levels, and a low transferrin saturation combined with high hepcidin levels and symptoms related to severe anemia, such as fatigue, palpitations, and exertional dyspnea [139]. The disease was discovered in a mask mouse model that showed iron refractory anemia and high serum hepcidin levels. It was demonstrated that mice with a null mutation on the TMPRSS6 gene have the same phenotype as humans carrying the TMPRSS6 mutation.

The prevalence of IRIDA is estimated to be <1/1,000,000. Although mutations are very rare, polymorphisms are more frequent, and some studies demonstrate that they can play a role in iron absorption, slowing the response to oral iron and aggravating symptoms in patients who are already at risk of developing iron deficiency, for example, celiac patients or fertile women [140].

The treatment regimen is iron supplementation; studies show that most patients need intravenous iron to improve hemoglobin levels. In many patients, transferrin saturation remains below the normal range despite iron treatment.

## 3. Conclusions

This review emphasizes that monitoring the iron status and applying therapeutic interventions is a valuable therapeutic approach that can be employed in combination with existing targeted drugs and immune-based therapies to enhance their efficacy.

Fundamental for the evaluation of iron metabolism is the analysis of iron parameters: serum iron, serum ferritin, and transferrin saturation associated with RBC parameters. Laboratory work-up allows for the discrimination and diagnosis of many iron-related diseases that can hardly be distinguished based on the clinical presentation alone. A deeper understanding of how a dysregulated cellular iron metabolism shapes the patient outcome could help to uncover new therapeutic avenues. In some instances, it might only be the shifting of iron between cellular compartments that affects disease progression and patient outcomes. The impressive progress made in the past years in understanding iron homeostasis in more detail is promising for the future treatment of iron overload diseases as well as iron deficiency. Several studies in the iron research field and clinical trials with new molecules are ongoing to fine-tune the treatment of iron-related disorders. 

## Figures and Tables

**Figure 1 pharmaceuticals-16-00329-f001:**
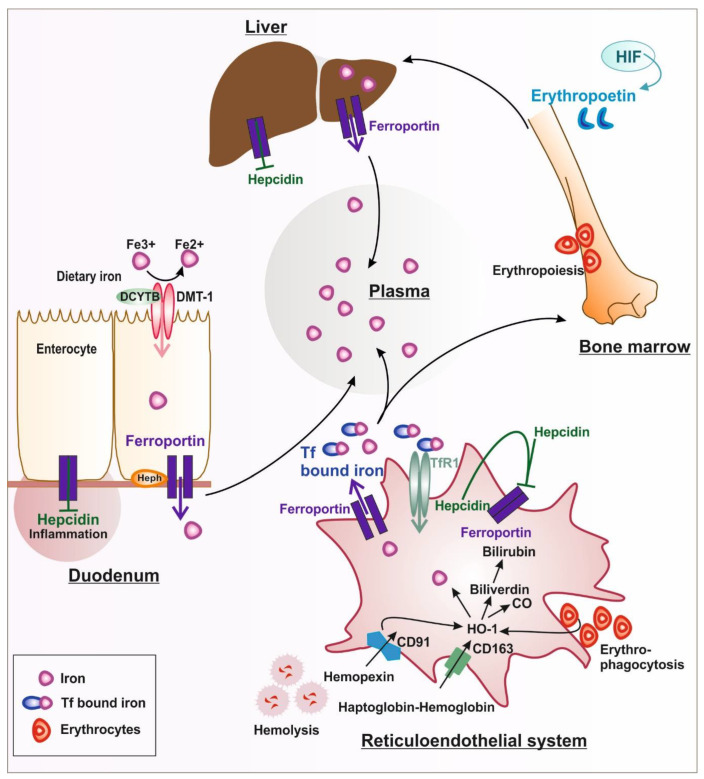
Main players in systemic iron metabolism.

**Figure 2 pharmaceuticals-16-00329-f002:**
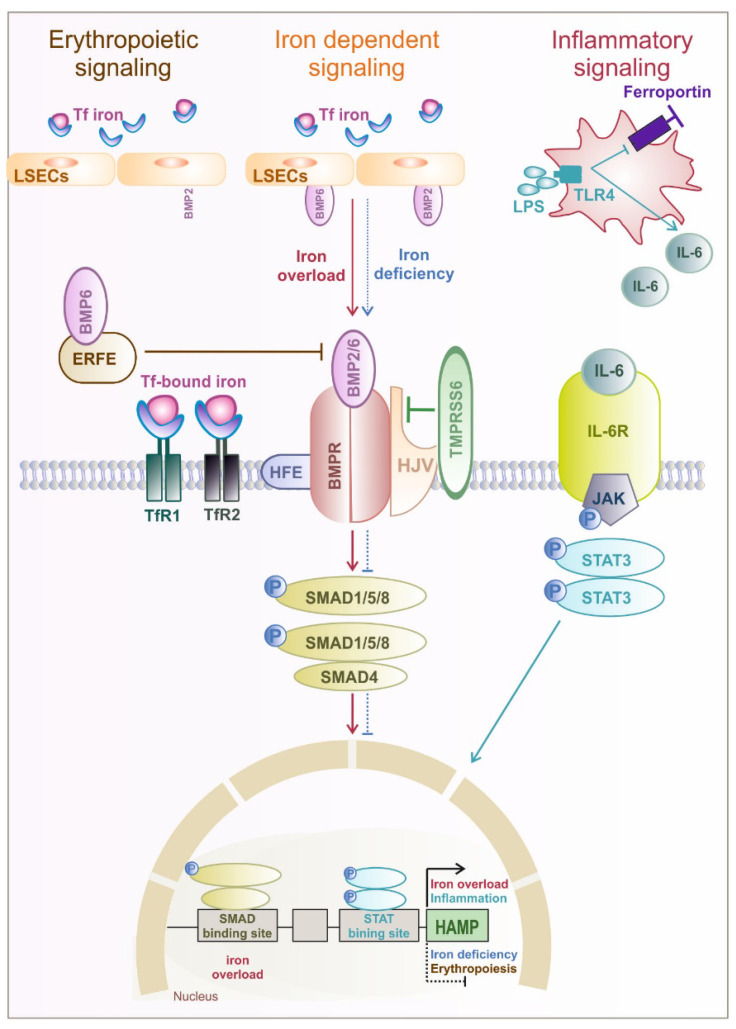
Mechanism regulating hepcidin expression.

**Table 1 pharmaceuticals-16-00329-t001:** Overview of iron parameters and hepcidin in the most common conditions of iron overload and iron deficiency. N: normal; ↓ reduced; ↑ elevated.

Disease			Serum Iron	Serum Ferritin	Transferrin Saturation	Hepcidin
Iron overload	Congenital	Hemochromatosis	↑	↑	↑	↓
		Thalassemia	↑	↑	↑	↓
		Sideroblastic anemia	↑	↑	↑	↓
	Acquired	Transfusion iron overload	↑	↑	↑	↑
Iron deficiency	Congenital	IRIDA	N/↓	N/↓	↓	↑
	Acquired	Absolute iron deficiency	N/↓	↓	↓	↓
		Functional iron deficiency	N	↑	↓	↑

**Table 2 pharmaceuticals-16-00329-t002:** New classification of hemochromatosis.

Novel Classification	Molecular Presentation
HFE-related	*C282Y*/*C282Y* *C282Y*/*other rare HFE pathogenic variants* H63D
Non-HFE-related	HJV-related (G320V, I222N, D249H, Q312X) HAMP-related (93delG, R56X, C70R, G71D) TFR2-related (V162del, A77D, Y250X, E60X) SLC40A1-related (A77D, I152F and L233P)
Digenic	HFE and/or non-HFE
Molecularly undefined	Molecular characterization not available

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
