# Peer review of "Interpreting Iron Homeostasis in Congenital and Acquired Disorders"

_pharmaceuticals, 2023, doi:10.3390/ph16030329_

Round 1

Reviewer 1 Report

The work presented by Scaramellini et al. is a review of iron metabolism in humans and the possible pathologies that may occur due to its alteration. Although it presents concepts of interest in the field, it does not provide new knowledge or new interpretations of the previous known. I suggest to integrate updated knowledge to draw new conclusions, so that it contributes something more to what has already been published (for example in doi: 10.3324/haematol.2019.232124). In addition, it would also be desirable to increase the focus of the review towards more novel or personalized therapies, in addition to review the classic ones.

Minor issues

There should be more connection between the different topics covered.

Some grammatical and punctuation errors need to be corrected.

In the case of congenital diseases it is necessary to include the MIM number of the disease (phenotype) and if possible of the gene (e.g. #235200 for hemochromatosis and #613609 for FH).

In the case of mutations referring to the protein, p. must always be used to name them (as in Table 2) and always using the same nomenclature (3 or 1 letter: always p.C282Y or always p.Cys282Tyr). In Table 2 some frequent mutations are missing, such as p.H63D for HFE and G71D for HAMP (among others).

In the case of HFE, the work said enough about the mouse, but little about the human phenotype.

Section 21.2.1 is underdeveloped, being excessively short.

Author Response

We thank the reviewer for the critical and very helpful feedback to our publication. We included new aspects into the review that add new knowledge to personalized therapies or therapies under development in the iron research field.  

Minor issues

There should be more connection between the different topics covered.

As suggested by the reviewer we connected the different parts of the review and added more knowledge about the interconnection of the different aspects.

Some grammatical and punctuation errors need to be corrected.

Corrected

In the case of congenital diseases it is necessary to include the MIM number of the disease (phenotype) and if possible of the gene (e.g. #235200 for hemochromatosis and #613609 for FH).

As mentioned by the reviewer we now included this information in the review.

In the case of mutations referring to the protein, p. must always be used to name them (as in Table 2) and always using the same nomenclature (3 or 1 letter: always p.C282Y or always p.Cys282Tyr). In Table 2 some frequent mutations are missing, such as p.H63D for HFE and G71D for HAMP (among others).

We thank the reviewer for this note and added the missing information in the table as well as in the text.

In the case of HFE, the work said enough about the mouse, but little about the human phenotype.

We included more information about the human phenotype of HFE. 

Section 21.2.1 is underdeveloped, being excessively short.

We added more information and details of chapter 2.1.2 in more depth explaining risk constellations and disease.

Reviewer 2 Report

Overall, it is a very well written manuscript with good effort from the Authors. There are some recommendations from me that I humbly believe would improve the manuscript:

1. Additional review on iron metabolism in cancers should be provide. Kindly add a subtopic and a few paragraphs. I believe this will be very interesting and add more utility for the article.

2. Also, the aspect of hepcidin in cancer should be added

3. Part 1.1 should be separated into several paragraphs for easier reading

4. Paragraph 1 should be combined with paragraph 2 in section 2.1.3

5. Mechanisms on how iron overload leads to heart failure and liver cirrhosis should be elaborated.

6. If time permits, a figure describing consequences of iron overload to organs and the mechanisms should be added.

7. The word “toa” should be change into “to a” in line 405

Author Response

We thank the reviewer for the positive feedback and recommendations. We included point by point answers to the different aspects.

  1. Additional review on iron metabolism in cancers should be provide. Kindly add a subtopic and a few paragraphs. I believe this will be very interesting and add more utility for the article.

Response: As suggested by the reviewer we added a subtopic about the iron metabolism in cancer. The chapter iron and cancer is now included as chapter 2.2 (page lines: 347-380).

  1. Also, the aspect of hepcidin in cancer should be added

Response: In the new subchapter about the iron metabolism of cancer we also included a part about hepcidin (page lines: 357-361).

  1. Part 1.1 should be separated into several paragraphs for easier reading

Response: In order, to address the reviewer’s comment, we subdivided part 1.1 into several paragraphs.

  1. Paragraph 1 should be combined with paragraph 2 in section 2.1.3

Response: We followed the reviewer’s suggestion and modified the paragraph accordingly.

  1. Mechanisms on how iron overload leads to heart failure and liver cirrhosis should be elaborated.

Response: We followed the reviewer’s suggestion and provided more information how iron overload leads to heart failure and liver cirrhosis (page lines: 315-320).

  1. If time permits, a figure describing consequences of iron overload to organs and the mechanisms should be added.

Response: We thank the reviewer for his/her suggestion, however, after weighing the options, we decided not to add a figure, since the topic was recently covered in other reviews, partly deviates from the objective of this review and last but not least, it would have required more time than that foreseen for the review deadline.

  1. The word “toa” should be change into “to a” in line 405

Response: Corrected.

Reviewer 3 Report

This review article is interesting, it provides a lot of informations about iron disorders in different types of anemia. It would be useful for authors to include in the review clinical characteristics of certain types of anemia mentioned in this article, such as symptomatology, severity of anemia, etc. Also, authors should include changes of MCV, MCH, MCHC for each type of described anemia. The conclusion is too general, it is necessary to write again new conclusion that will be linked in better way with this review article.

Author Response

We thank the reviewer for the positive feedback and recommendations. We included point by point answers to the different aspects.

Response: According to the reviewer’s comments, we added the suggested information in the review

The conclusion is too general, it is necessary to write again new conclusion that will be linked in better way with this review article.

Response: As suggested by the reviewer, we rewrote the conclusion and linked it more to the review article.

Reviewer 4 Report

The authors reviewed current knowledge regarding iron metabolism and conditions, either congenital or acquired, associated with iron overload or depletion. The manuscript is well organized and the explanation of the involved mechanisms is accompanied by eloquent summary figures.

However, some improvements are recommended.

Row 33: I propose: ...ferric iron (Fe+3), therefore dietary iron first needs to be reduced in the duodenal lumen to ferrous iron (Fe+2) before it can be... This text and figure 1 are linked. Please revise figure 1 as it shows iron oxidation and not iron reduction (at the level of the apical membrane).  

In section 2.2.1 the authors discuss the available iron formulations. However, new and performant oral drugs are now available and I recommend to be highlighted (sucrosomial iron and liposomal iron). Since there is evidence that oral sucrosomial iron is of comparable efficacy to iv iron, the indications for iv iron preparations should be discussed in this context.

The authors should emphasize what this study brings in addition to the already published literature (the strengths of the study).

The Conclusions section needs revision for the following reasons:

1. Text repetition: „difficult to diagnose and differentiate only based on clinical presentation” and „hardly be distinguished based on the clinical presentation alone”.

2. information not discussed in the text appears for the first time in the conclusions: Many open questions remain about the role of iron (deficiency or excess) in the process of atherosclerosis, ... neurodegenerative disorders.

Spell check required: DCYTb or DCYTB ? TMPRSS6 or Tmprss6 ?

Thank you!

Author Response

We thank the reviewer for the positive feedback and recommendations. We included point by point answers to the different aspects.

Row 33: I propose: ...ferric iron (Fe+3), therefore dietary iron first needs to be reduced in the duodenal lumen to ferrous iron (Fe+2) before it can be... This text and figure 1 are linked. Please revise figure 1 as it shows iron oxidation and not iron reduction (at the level of the apical membrane).  

Response: We thank the reviewer for this critical comment and changed the text and figure according to the suggestion.

In section 2.2.1 the authors discuss the available iron formulations. However, new and performant oral drugs are now available and I recommend to be highlighted (sucrosomial iron and liposomal iron). Since there is evidence that oral sucrosomial iron is of comparable efficacy to iv iron, the indications for iv iron preparations should be discussed in this context.

The authors should emphasize what this study brings in addition to the already published literature (the strengths of the study).

Response: As suggested by the reviewer, we added a small paragraph about liposomal iron.

The Conclusions section needs revision for the following reasons:

  1. Text repetition: „difficult to diagnose and differentiate only based on clinical presentation” and „hardly be distinguished based on the clinical presentation alone”.
  2. information not discussed in the text appears for the first time in the conclusions: Many open questions remain about the role of iron (deficiency or excess) in the process of atherosclerosis, ... neurodegenerative disorders.

Response: As suggested by the reviewer we revised the conclusion part.

Spell check required: DCYTb or DCYTB ? TMPRSS6 or Tmprss6 ?

Response: Corrected.

Round 2

Reviewer 1 Report

Major issues have not been fixed or even considered. Nor have the minor ones been fully resolved; f.i. the p. before de mutation (protein) is still all over in the text and the MIM numbers have not been added.

Author Response

We thank the reviewer for giving us another chance to improve the quality of our article and for another chance to invested sufficiently time to implement the valuable suggestions.

We now introduced more new therapeutic perspectives by adding, for instance, hints about new therapies for hemochromatosis and the use of liposomal iron for treating iron deficiency

However, to not dive too much on therapies and rather stay with pathophysiology, we left it with this. We feel that our review now gives an overview on the role of iron in iron-related diseases and additionally highlights the therapeutic importance of balancing iron metabolism.

We are very sorry that in the first revision we missed to remove “p.” in some parts of the article especially in table. 2. Changes are now highlighted in green. In addition, we also added the missing MIM numbers.

Reviewer 4 Report

The manuscript was revised with great attention and seriousness. The authors took into account all the recommendations made and, starting from them, made valuable interventions in the text and made major improvements to the manuscript.

I think that the result of the authors' effort is a very good one.

I would like to highlight a few aspects:

·         the abstract has been polished and improved its ability to attract the attention of readers.

·         iron homeostasis in patients with neoplasia is widely debated and is a modern topic of wide interest.

·         the conclusions better summarize the valuable elements of the manuscript.

Very small observation: Row 256. Please revise. I think the intended meaning was "admitted".

Thank you!

Author Response

We thank the reviewer again for the giving us precious feedback and for acknowledging our efforts to improve the publication.